# Feeding Behavior Comparison of Bean Bugs, *Riptortus pedestris* and *Halyomorpha halys* on Different Soybean Cultivars

**DOI:** 10.3390/insects14040322

**Published:** 2023-03-27

**Authors:** Seong-Bin Park, Hyun-Na Koo, Seung-Ju Seok, Hyun-Kyung Kim, Hwi-Jong Yi, Gil-Hah Kim

**Affiliations:** 1Department of Plant Medicine, Chungbuk National University, Cheongju 28644, Republic of Korea; 2Department of Southern Area Crop Science, National Institute of Crop Science, Rural Development Administration, Miryang 50424, Republic of Korea

**Keywords:** electrical penetration graph, feeding preference, feeding wave, seed damage

## Abstract

**Simple Summary:**

This study compared the feeding behavior of *Riptortus pedestris* and *Halyomorpha halys* on six soybean cultivars. In addition, the damage caused by *R. pedestris* and *H. halys* was field-surveyed. The non-penetration waveform was the shortest in Pungsannamul and the longest in Daepung-2ho. In the case of feeding waveforms of the xylem and phloem, it was the longest in Pungsannamul and the shortest in Daepung-2ho, respectively. When six soybean cultivars were planted in two fields, and the seed damage caused by hemipterans was investigated; in both fields, the proportions of damage type B and damage type C were the highest in Pungsannamul and the lowest in Daepung-2ho. The above results will be used as basic information in the control of hemipteran insect pests.

**Abstract:**

*Riptortus pedestris* (Fabricius) and *Halyomorpha halys* (Stål) are the major pests that feed on soybean pods, seeds, and fruits. Higher populations and damage occur during the soybean maturity stages (podding to harvest). To compare the feeding behavior of *R. pedestris* and *H. halys*, we used the six most cultivated cultivars (Daepung-2ho, Daechan, Pungsannamul, Daewon, Seonpung, and Seoritae) in Korea using the electropenetrography (EPG) technique. Both *R. pedestris* and *H. halys*, the NP (non-penetration), a non-probing waveform, was the shortest in the Pungsannamul (298 and 268 min) and the longest in the Daepung-2ho (334 and 339 min), respectively. The feeding waveforms Pb (phloem feeding: E1-Salivation and E2-Sap feeding) and G (xylem feeding) were the longest in Pungsannamul and the shortest in Daepung-2ho. In addition, as a result of investigating the damage rate by planting six cultivars of beans in the field, as expected, the proportions of damage types B and C were highest in Pungsannamul and lowest in Daepung-2ho. These results reveal that both bug species ingest xylem sap from leaflets and stems using a salivary sheath strategy to acquire water and nutrients from soybean pods/seeds with cell-rupture tactics. This study provides perceptive information to understand the feeding behavior, field occurrence, and damage patterns of *R. pedestris* and *H. halys*, which may have key implications for the management of hemipteran pests by determining the specificity and susceptibility of host plants.

## 1. Introduction

Hemipterans, particularly Alydidae, and Pentatomids, have been considered the most damaging pests of several legumes, cereals, and fruits in Korea, Japan, China, and India [1,2,3,4]. Among hemipterans, *R. pedestris* (Hemiptera: Alydidae) and *H. halys* (Hemiptera: Pentatomidae) are major species that occur in East and South Asian countries, including Korea [1,5,6,7,8]. The population of bug species (especially *R. pedestris* and *H. halys*) is spreading across Korea from south to north; bug species have been found on soybean (*Glycine max*), sesame (*Sesamum indicum*), sorghum (*Sorghum bicolor*), proso millet (*Panicum miliaceum*), fox-tail millet (*Setaria* italica), kiwifruit, etc. [9,10,11,12,13]. Large populations of *R. pedestris* and *H. halys* are commonly observed in soybean fields at the soybean maturity stage (pod filling to harvest) in Korea and Japan [14,15]. Nymphs and adult bugs damage pods and seeds, and fruits by piercing them and sucking from them with needle-like mouthparts [1,6,10]. Damage from both nymphs and adults impacts seed quality [16] and reduces germination potential and yield [17,18].

Conventional broad-spectrum insecticides are widely used to control bugs [19,20,21]. However, in practical terms, managing bugs is difficult because of their high mobility and insecticide-avoidance behavior [22]. To achieve proper management, growers frequently apply broad-spectrum insecticides [7,23,24]. Heavy and frequent insecticide application greatly impacts the abundance of natural enemies, leads to secondary pest outbreaks, and severely disrupts integrated pest management programs [20,25]. Therefore, developing and cultivating insect-resistant soybean varieties are effective ways to reduce the damage from stink bugs. Hemiptera pests feed on the phloem of host plants via a long needle [16]. Hemipterans secrete water-soluble and coagulant saliva when sucking plants [26]. They explore the epidermis between epidermal cells and phloem tissues to find a suitable part for extracting nutrients [27]. As such, it is not easy to study the feeding behavior of bugs because they simultaneously perform actions, such as salivation, exploration, and sucking [28,29]. Electrical penetration graphs or electropenetrography (EPG) is considered a leading technique that is widely employed to examine/evaluate the feeding behaviors of piercing-sucking insects [30]. The feeding behavior of insects, such as aphid species can be analyzed using DC (direct current)-EPG [31]. With these techniques, it has become possible to analyze the salivation, search, and suction patterns of bugs in plant tissues. The feeding behavior of aphids was the first activity to be studied with EPG [32]. Since then, EPG has been widely used to study the feeding behavior of piercing–sucking-type insects, such as locusts, whiteflies, leaf beetles [33], and tree lice [34]. However, there are limited studies relative to the feeding behavior of bugs using EPG [35].

Soybean (*Glycine max* (L.) Merrill) is one of the most important crops in the world and in Korea as well. Soybean is cultivated in large areas of arable land worldwide. It is generally known that the distribution of the wild soybean, which is the ancestor of the currently cultivated soybean, is limited to the East Asia regions, including China, Korea, and Japan. Soybean production in South Korea has gradually decreased during the past 40 years; however, its importance as a major crop has not diminished, probably because soybean had a long history of cultivation intimately associated with the traditional food culture of South Korea. Soybeans, such as Daewon, Daechan, Daepung-2ho, Seonpung, Pungsannamul, and Seoritae, are popular Korean cultivars and have a higher concentration of soyasaponin and isoflavone. They are widely cultivated in South Korea due to their plant disease-resistance characteristics [36]. However, whether they are insect-pest-resistant is not well known. Therefore, in this study, the level of insect resistance of soybean cultivars against *R. pedestris* and *H. halys* was analyzed by investigating the feeding behavior using the EPG system. In addition, the damage intensity of *R. pedestris* and *H. halys* under field conditions was assessed.

## 2. Materials and Methods

### 2.1. Insect Rearing

The bean bug *R. pedestris* and *H. halys* were collected in soybean fields, in Cheongju, South Korea, in 2020. They were reared in the Insect Toxicology Laboratory of Chungbuk National University (South Korea) and placed in acrylic rearing cages (40 × 40 × 40 cm) with side ventilation. Soybean plants grown in a glasshouse and soybean seeds were provided as food sources. Insects were reared at 27 ± 2 °C, 50–55% RH, and a 16:8 h; L:D photoperiod. Newly emerged adults of both stink bug species were collected and transferred to separate cages.

### 2.2. EPG Recordings

The probing and feeding behaviors of *R. pedestris* and *H. halys* were recorded using the DC-EPG Systems with 1 GΩ of resistance (Wageningen, Güeldres, The Netherlands) by Tjallingii [31,37]. Second instar nymphs (4 hours starved) of both species were used in the experiment. Gold wire (AlfaAesar, Waltham, MA, USA) with a diameter of 50 μm and a length of 3–4 cm was attached to the center of the chest and dorsal side of each insect using a dissecting microscope (Olympus Co., Tokyo, Japan). For attachment, conductive electroconductive silver paste (P-100, JP) was used as an adhesive. Insects with gold wires were then connected to the Giga-8 DC EPG amplifier (EPG Systems, Wageningen, The Netherlands). They were then placed into bean stems (14 days old) of each soybean cultivar, and the output voltage was recorded for 6 hours. The voltage changed due to the difference in electrical resistance resulting from the stylet penetration movement when the stylet of an insect was inserted into the plant. Stink bugs were tested individually, using a new adult for each replicate. The recording was repeated 49–71 times for each soybean cultivar. All experiments were performed in a Faraday cage surrounded by copper mesh to minimize noise disturbing electrical signal recording.

### 2.3. Analysis of EPG Waveforms

The total duration of EPG waveforms for each soybean cultivar was measured. EPG signals were acquired and analyzed using Stylet+ software for Windows (EPG Systems). EPG variables were processed using the EPG-Excel Data Workbook developed by Sarriá et al. (2009) [38]. The electrical signals and their correlations with stink bug behavior were scored based on the categories described by Tjallingii (1990): non-probing (stylets external to the plant, NP), xylem ingestion (waveform G), and phloem activities (pooling together waveforms E1 and E2 reflecting salivation into sieve elements and phloem ingestion, respectively). 

To compare the probing behavior of *R. pedestris* and *H. halys* on soybean leaves, a selected set of EPG variables was calculated as follows: the total waveform duration for each insect (the total duration of a waveform summed over all occurrences of the waveform for each insect) and the mean duration of waveform events for each insect (the total waveform duration divided by the number of waveform events for each insect). Then, the data of the total duration of the waveforms were used to calculate the mean percentage of time spent on each stylet activity (NP, G, E1, and E2) per soybean cultivar.

### 2.4. Field Experiment

On 30 May 2021, two fields (CJ-1 and CJ-2) of soybean cultivation were established on a farm located at Chungbuk National University, Cheongju-si, Chungbuk province, South Korea. Six soybean cultivars (Daepung-2ho, Daechan, Pungsannamul, Daewon, Seonpung, and Seoritae) recommended by the Korea Rural Development Administration (Jeonju, South Korea) were sown. The test plots were arranged in a randomized block design with 6 treatments and 3 replications. Two beans were sown in one row at an interval of 60 × 15 cm in each test plot. To check whether they were being damaged by the stink bug, the level of stink bug occurrence was investigated at 7-day intervals. The survey was conducted from August 31 to October 19 (grain filling period), the date when the stink bug moves to the soybean field. We walked slowly through the soybean fields, “searching and netting” the insect pests. Succulent insects other than stink bugs were not found; only a few moth larvae were found, but they were removed.

### 2.5. Damage Measurement and Analysis

The seed damage rate was determined by the soybean development stage classification method [39] and the Criteria for Research and Analysis of Agricultural Science and Technology [40]. Harvested soybeans were peeled off all pods, and the damage to the seeds was visually inspected. Seed damage was reported according to the classification criteria developed by Jung et al. [41] (Table 1): type A: normal seeds without feeding marks; type B: seeds that are normal in shape but show feeding marks on the surface; type C: seeds that are deformed in size or shape; and type D: seeds that are immature and have little shape. The number of pods and seeds harvested for each cultivar was investigated. The ratio was also analyzed for each type.

### 2.6. Data Analysis

The two sting bugs were studied independently. First, the raw data of *R. pedestris* and *H. halys* were checked for normality and homogeneity of variance using Shapiro–Wilk W and were transformed with sqrt (x + 1) and ln (x + 1) if needed. According to a previous report by Vázquez et al. (2020) [42], most EPG variables did not follow a normal distribution; therefore, they were compared by Student’s *t*-test for parametric variables at a 0.05 significance level using SPSS V26.0 Statistics software (IBM^®^) for *R. pedestris* data. Additionally, the comparisons between the proportions of individuals that produced a given type of wave were analyzed using Tukey’s HSD (*p* < 0.05) [43]. To determine the preference of soybean cultivars for stink bug probing and feeding behaviors, non-parametric analyses were performed using Tukey’s test. In addition, the seed damage rate by stink bugs was also analyzed using Tukey’s HSD (*p* < 0.05).

## 3. Results

### 3.1. The EPG Waveforms of R. pedestris and H. halys on Six Soybean Cultivars

The changes in the feeding behavior of the *R. pedestris* and *H. halys* were analyzed as waveforms (voltage changes due to the differences in electrical resistance) using the EPG system. Figure 1a shows the entire waveform measured 1 hour after the stink bug was placed on the host plant. A close analysis of the entire waveform shows the NP (non-penetration) waveform, as shown in Figure 1b. NP is a non-probe waveform for which the stylet and the host surface are completely separated. It is the waveform shown when the stink bug is resting before or after eating its host or when it does not prefer its plant. G (Xylem phase) is the waveform when the stink bug feeds on the xylem of the plant (Figure 1c). The phloem phase (when the stink bug feeds on the host phloem) was analyzed by dividing it into E1 (Salivation), which secretes saliva, and E2 (Sap feeding), which sucks the phloem (Figure 1d).

After recognizing each waveform, we compared and analyzed the feeding behavior of two species of stink bugs on six soybean cultivars (Table 2) for 6 hours. The response variable values were the total waveform duration (min) of NP, G, E1, and E2. For both stink bugs, the time spent feeding for 6 hours in the six soybean cultivars was the longest for the NP without stylet insertion, and the time spent for phloem feeding (E1, E2) was shorter than for xylem feeding (G). First, the results revealed that the duration of the NP waveforms of *R. pedestris* (332.47 ± 10.8 min) was longer than that of *H. halys* (303.83 ± 40.2 min) only in the Daechan cultivar (*p* = 0.0095). The G waveform did not show any significance between the *R. pedestris* and *H. halys* in any of the six soybean cultivars. The duration of the E1 waveform of *R. pedestris* was longer in Daepung-2ho (*p* < 0.0001), Daechan (*p* = 0.0273), Seonpung (*p* < 0.0001), and Seoritae (*p* < 0.0001) than that of *H. halys*. In the E2 waveform, the time of *R. pedestris* was significantly longer than that of *H. halys* in Daepung-2ho (9.84 ± 8.9 min) (*p* < 0.0001) and Seoritae (16.82 ± 13.1 min) (*p* < 0.0001). However, *H. halys* showed the E2 waveform for 0 min in Daepung-2ho and Seoritae, unlike *R. pedestris*.

The percentage of time that sting bugs spent conducting each probing activity is represented in Table 3. When *R. pedestris* was placed on Pungsannamul, the percentage of the total time spent on non-probing events (np) was shorter (82.71%) when compared with that on Daepung-2ho (92.97%) (*p* = 0.0001). Additionally, the percentage of time spent on phloem ingestion (E2) was significantly (*p* = 0.001) lower on Daepung-2ho (2.73%) than on Pungsannamul (7.91%). In other words, this result indicates that *R. pedestris* prefers Pungsannamul to Daepung-2ho. No differences were found between treatments for salivation in phloem sieve elements (E1) (*p* = 0.147). In *H. halys*, NP was the shortest (74.55%) in Pungsannamul and the longest (94.23%) in Daepung-2ho (*p* = 0.0001), similar to that of the *R. pedestris*. Additionally, E2 was the longest (11.2%) in Pungsannamul and 0% in Daepoong-2ho and Seoritae (*p* = 0.023). These results indicate that *H. halys* does not prefer Daepung-2ho and Seoritae cultivars.

### 3.2. Occurrence Density of Stink Bugs in Soybean Fields

First, to analyze the occurrence density of stink bugs in a soybean field, we planted six soybean cultivars (Daepung-2ho, Daechan, Pungsannamul, Daewon, Seonpung, and Seoritae) in fields A and B of Cheongju, Chungbuk province. The distribution of the stink bugs in the soybean fields was investigated for approximately one month (31 Aug–19 Oct) (Figure 2). To survey, we walked slowly through the soybean fields (CJ-1 and CJ-2), “searching and netting” the insect pests. Succulent insects other than stink bugs were not found; only a few moth larvae were found, but they were removed. As a result, six species (*R. pedestris*, *H. halys*, *Nezara antennata*, *Homalogonia obtusa*, *Dolycoris baccarum*, and *Acanthocoris sordiduswere*) were mainly caught, and the number was the highest in September. In both fields, the dominant species was *R. pedestris* (A, total No. 347 and B, 201), followed by *H. halys* (A, total No. 198 and B, 120) and *N. antennata*. (A, total No. 180 and B, 88).

### 3.3. Seed Damage Assessment

The number of seeds compared to the yield of soybean pods ranged from 1.91 (Seoritae, 209.3:399.3) to 2.22 (Daepung-2ho, 206.3:457.3), with Daepung-2ho showing the highest value (Table 4). Soybean seed damage types (types B and C and combined (B + C)) were found to vary according to soybean cultivars. The number of type-B and combined (B + C) seeds was significantly different among soybean cultivars in field CJ-1 (F = 19.56, df = 5, *p* < 0.0001 for type B, and F = 11.60, df = 5, *p* = 0.0003 for combined type); however, type-C seeds were not different among soybean cultivars (F = 2.48, df = 5, *p* = 0.092). The least damaged soybean cultivar was Daepung-2ho (17.9%), while the most damaged soybean cultivar was Pungsannamul (53.5%). The pattern of the seed damage rate of soybean cultivars in field CJ-2 was the same as that in field CJ-1. The numbers of type-B and combined (B + C) seeds were significantly different among soybean cultivars (F = 11.93, df = 5, *p* = 0.0003 for type B, and F = 9.01, df = 5, *p* = 0.0009 for combined type); however, type-C seeds were not different among soybean cultivars (F = 1.75, df = 5, *p* = 0.199). The least damage was recorded for Daepung-2ho (22.2%), and the most damage was recorded for Pungsannamul (45.7%). Therefore, Daepung-2ho can reduce damage by stink bugs and is more resistant than other soybean cultivars.

## 4. Discussion

In this study, we compared the feeding behavior of *R. pedestris* and *H. halys* on six soybean cultivars. The EPG analysis revealed that the NP waveform was the shortest in Pungsannamul and the longest in Daepung-2ho. In the case of feeding waveforms of the xylem and phloem, it was the longest in Pungsannamul and the shortest in Daepung-2ho. From these results, it seems that the two stink bugs prefer Daepung-2ho less than the Pungsannamul cultivar. By examining the total feeding pattern recording time and non-probe waveform recording time for each soybean cultivar, it is possible to know the bean cultivars preferred by the stink bug [44,45]. In general, insects of Hemiptera use two feeding strategies [26]. One is the salivary sheath strategy, which gels saliva around the stylet to form an envelope. The other is cell rupture [30,46]. It secretes saliva and destroys cells while moving in and out after inserting stylet into host tissue [30,47,48]. In addition, cell rupture is divided into the salivation phase and the feeding phase. Stink bugs use only one strategy or both, depending on the species [49]. The two species of stink bug used in this experiment showed a pattern of dividing the salivation phase and the feeding phase, so they seem to use cell rupture as a feeding strategy. In the process of EPG analysis, the sucking of xylem by stink bugs was performed only once, and no more than two suckings were observed during the study period. The feeding pattern we recorded in this study is in line with that of Lucini et al. [35], who also found a similar pattern in *Piezodorus guildinii*. Sucking the xylem is considered a strategy used by Aphidoidea and Psyllidae to avoid dehydration and maintain water balance [31,50,51]. Similarly, in both stink bugs, sucking the xylem seems to be a means to avoid dehydration and maintain water balance, so it is thought that they did not suck the xylem more than two times. Interestingly, no waveforms of E1 and E2 were observed in Daepung-2ho and Seoritae. In the results of the development and reproduction experiment by soybean cultivar, the development period was slow and the survival rate was low in Daepung-2ho and Seoritae (data not shown). In addition, analyzing the total duration of G, E1, and E2 waveform, except for Daepung-2ho and Seoritae, they feed more xylem but feed similarly to sap. In the case of feeding only the xylem, it seems that the stink bug did not eat the phloem because it is not a preferred soybean cultivar. It is possible that the longer the experiment time, the higher the sap-feeding rate. However, since the EPG experiment was conducted with highly active insects, they may have been under a lot of stress. When comparing the feeding waveform of stink bugs with the feeding waveform of cotton aphids (*Aphis gossypii* Glover, Hemiptera: Aphididae) [52], we recorded a threefold higher NP time for stink bugs than for cotton aphids. In our previous study, the feeding pattern of the NP waveform in cotton aphids is usually 70–80 min, the G waveform is about 20 min, the E1 waveform is about 25 min, and the E2 waveform is about 200 min. Interestingly, resistant cotton aphids were found to be more active in finding suitable feeding sites on insecticide-treated hosts than susceptible cotton aphids. Therefore, the feeding patterns of stink bugs and cotton aphids were different. Our study also shows that there are differences among stink bugs. In addition, Daepung-2ho and Seoritae can be resistant cultivars against *H. halys*. Overall, for both stink bugs, the feeding time was the shortest in Daepung-2ho and the longest in Pungsannamul. In the other four soybean cultivars, the feeding patterns were different according to the stink bug species. Host plant suitability and feeding preferences of insects are determined by various factors such as taste, smell, leaf color, leaf hair, and leaf thickness.

The dominant species of the stink bug on six soybean cultivars were *R. pedestris*, followed by *H. halys* and *N. antennata*. The number was the highest in September. In addition, the least damaged soybean cultivar was Daepung-2ho (17.9%, 22.2 %), while the most damaged soybean cultivar was Pungsannamul (53.5%, 45.7%) in fields A and B, respectively. The number of seeds compared to the yield of soybean pods, Daepung-2ho showed the highest value. Daepung-2ho is currently a cultivar of the highest yield (345 Kg/10a) in South Korea. This cultivar is also tolerant to lodging, fire blight, and seed shattering. Therefore, Daepung-2ho has a good yield and can reduce damage to stink bugs.

EPG waveform analysis also confirmed that Pungsannamul had a high preference for stink bug feeding. However, the longer feeding time of stink bugs and the higher damage rate of soybeans did not necessarily indicate the preference of stink bugs. In the EPG experiment, it may be that the nutritional level of soybean could not be satisfied, so it needs to be fed for a long time. In the field experiment, it may be that soybeans are less resistant due to being more responsive to feeding or because of long-time feeding. The findings of this study are similar to those of previous studies showing that stink bug damage varies according to soybean cultivar [10,53]. Oh et al. [54], however, reported that Pungsannamul has less stink bug pod damage but higher yield reduction, which contradicts our estimates. This discrepancy may be due to the availability of other palatable plant hosts and crops/fruits near soybean fields planted with Pungsannamul and biotic environmental variability. A previous study indicated that plant species and genotype and abiotic factors (humidity, temperature, and photoperiod) are significant factors influencing the cultivar of volatiles released by host plants [55]. In this experiment, the stink bugs showed significantly higher preferences toward the Pungsannamul soybean cultivar and appeared to prefer the Pungsannamul bean the most, findings that contradict those of Choi et al. [6], who suggested that Pungsannamul would have insect-resistant characteristics when considering the growth and weight change of stink bugs. This discrepancy might be influenced by the passage of time or environmental changes.

In conclusion, both *R. pedestris* and *H. halys* did not prefer Daepung-2ho as food, among the six soybean cultivars, and there was little damage from them. Further studies need to determine the effects on the growth, survival rate, and oviposition numbers of *R. pedestris* and *H. halys*. Therefore, we are currently conducting research in our laboratory.

## Figures and Tables

**Figure 1 insects-14-00322-f001:**
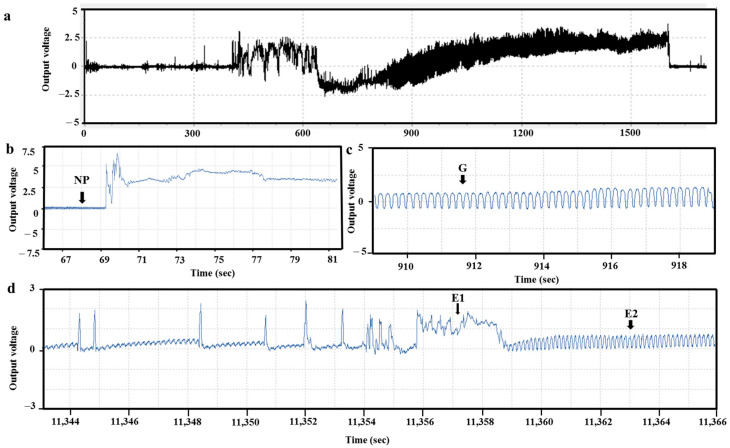
Typical EPG waveform of *R. pedestris* and *H. halys* during 1 h of recording and characteristic waveforms (panel (**a**)). Panel (**b**) is the NP waveform (non-penetration); panel (**c**) is the G waveform (Xylem feeding phase), and panel (**d**) is the E1 (Salivation) and E2 waveform (Sap feeding phase). The y-axis represents voltage, and the x-axis represents time (s).

**Figure 2 insects-14-00322-f002:**
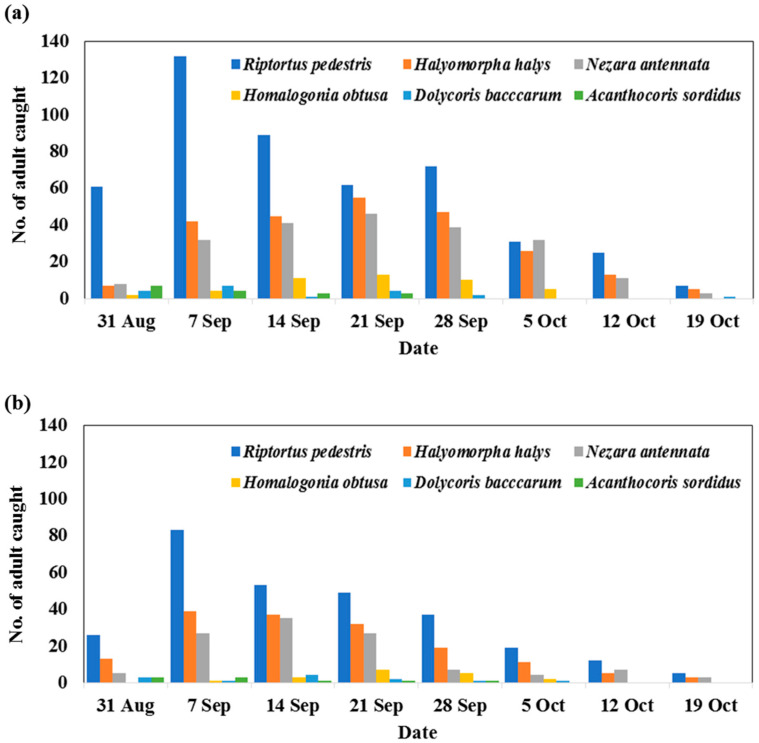
Change of density of stink bugs observed in soybean fields from 31 Aug to 19 Oct in Cheongju, Chungbuk province in 2021. Panel (**a**) is soybean field CJ-1 and panel (**b**) is soybean field CJ-2.

**Table 1 insects-14-00322-t001:** Classification of injury types of soybean seeds caused by hemipteran bugs.

Type	Injury Phenomena
A seed	Normal shape, non-injury
B seed	Normal shape but distinct injury marks
C seed	Deformed shape or diseased seed
D seed	Immature and undeveloped seed

**Table 2 insects-14-00322-t002:** EPG variable values (means ± standard error) during the probing and feeding behavior of *R. pedestris* and *H. halys* on six soybean cultivars during a 6-hour recording.

Soybean Cultivar	Total Waveform Duration (min)											
	Non-Probing, NP			Xylem Phase, G			Phloem Phase					
							E1			E2		
	*R. pedestris*	*H. halys*	*p*	*R. pedestris*	*H. halys*	*p*	*R. pedestris*	*H. halys*	*p*	*R. pedestris*	*H. halys*	*p*
Daepung-2ho	334.92 ± 10.8 a	339.24 ± 15.0 a	0.683	14.55 ± 11.9 a	20.76 ± 15.0 a	0.388	0.93 ± 0.04 a	0.0 ± 0.0 b	**<0.0001**	9.84 ± 8.9 a	0.0 ± 0.0 b	**<0.0001**
Daechan	332.47 ± 13.7 a	303.83 ± 40.2 b	**0.0095**	15.64 ± 10.8 a	40.84 ± 39.5 a	0.057	1.27 ± 0.17 a	0.02 ± 0.0 b	**0.0273**	11.09 ± 9.4 a	15.31 ± 13.3 a	0.5516
Pungsannamul	298.12 ± 29.1 a	268.38 ± 32.8 a	0.6654	32.12 ± 13.2 a	50.67 ± 12.2 a	0.668	1.69 ± 0.23 a	0.62 ± 0.4 a	0.2444	28.50 ± 22.9 a	40.33 ± 29.3 a	0.4962
Daewon	333.55 ± 13.2 a	306.47 ± 18.4 a	0.3346	14.44 ± 12.5 a	37.45 ± 24.8 a	0.244	1.01 ± 0.16 a	0.37 ± 0.2 a	0.5678	11.53 ± 10.8 a	15.71 ± 11.2 a	0.7483
Seonpung	321.42 ± 20.2 a	323.16 ± 23.6 a	0.7235	22.79 ± 20.4 a	25.20 ± 24.5 a	0.425	1.25 ± 0.35 a	0.24 ± 0.0 b	**<0.0001**	14.91 ± 10.6 a	11.56 ± 9.2 a	0.7728
Seoritae	321.36 ± 23.4 a	332.24 ± 14.8 a	0.7373	20.94 ± 18.1 a	27.76 ± 14.8 a	0.902	1.31 ± 0.12 a	0.0 ± 0.0 b	**<0.0001**	16.82 ± 13.1 a	0.0 ± 0.0 b	**<0.0001**

Means followed by the same letter (lowercase), in the same row, do not differ significantly (*p* > 0.05) using Student’s *t*-test. Highlighted in bold are those variables that showed striking significant differences between insect species.

**Table 3 insects-14-00322-t003:** Total duration of each waveform event (NP, G, E1, and E2) as proportions of the time spent during the 6 hours of EPG recording of *R. pedestris* and *H. halys*. Highlighted in bold are those variables that showed striking significant differences between soybean cultivars.

Species	Behavior *	Soybean Cultivar						*p*
		Daepung-2ho	Daechan	Pungsannamul	Daewon	Seonpung	Seoritae	
*R*. *pedestris*	Non-probing (NP)	92.97 a	92.23 ab	82.71 b	92.52 a	89.19 ab	89.16 ab	**0.0001**
	Xylem phase G	4.04 c	4.34 c	8.91 a	4.01 c	6.32 ab	5.81 b	**0.0018**
	Salivation into Phloem (E1)	0.26 a	0.35 a	0.47 a	0.28 a	0.35 a	0.36 a	0.147
	Phloem ingestion (E2)	2.73 b	3.08 b	7.91 a	3.20 b	4.14 ab	4.67 ab	**0.001**
*H*. *halys*	Non-probing (NP)	94.23 a	84.40 cd	74.55 d	85.13 cd	89.73 bc	92.29 ab	**0.0001**
	Xylem phase G	5.77 c	11.34 ab	14.08 a	10.40 ab	7.00 bc	7.71 bc	**0.0086**
	Salivation into Phloem (E1)	0.00 d	0.01 d	0.17 a	0.10 b	0.07 c	0.00 d	**0.0004**
	Phloem ingestion (E2)	0.00 c	4.25 ab	11.20 a	4.36 ab	3.21 b	0.00 c	**0.023**

* Variables followed by the same letter, in the same row, do not differ significantly (*p* > 0.05) according to Tukey’s HSD. Highlighted in bold are those variables that showed striking significant differences between insect species.

**Table 4 insects-14-00322-t004:** Comparison of injured soybean seeds and injury type caused by stink bugs on six soybean cultivars.

Site	Bean Cultivar	No. of Total Pod	No. of Total Seed	Rate of B-Type Seed	Rate of C-Type Seed	Damage Rate (B + C)/Total Seed
A	Daepung 2ho	206.3 ± 15.1	457.3 ± 13.6	5.8 ± 4.1 c	12.1 ± 3.5 a	17.9 ± 5.6 c
	Daechan	208.3 ± 7.2	458.3 ± 28.4	12.6 ± 2.6 bc	18.7 ± 2.1 a	31.3 ± 4.7 bc
	Pungsannamul	204.7 ± 8.1	429.0 ± 46.9	34.4 ± 5.0 a	19.1 ± 0.5 a	53.5 ± 4.9 a
	Daewon	202.7 ± 11.5	392.3 ± 27.4	17.6 ± 0.5 b	17.2 ± 7.3 a	34.8 ± 6.8 b
	Seonpung	194.7 ± 24.9	407.3 ± 38.0	15.3 ± 5.6 bc	16.2 ± 3.2 a	31.5 ± 8.2 bc
	Seoritae	209.3 ± 9.0	399.3 ± 34.1	15.4 ± 2.0 bc	10.5 ± 3.5 a	26.0 ± 5.1 bc
	*p* value			0.0001	0.092	0.0003
	F value			19.56	2.48	11.60
	df			5	5	5
B	Daepung 2ho	203.7 ± 13.8	459.3 ± 47.4	15.2 ± 4.6 b	7.0 ± 1.4 a	22.2 ± 4.7 b
	Daechan	212.7 ± 7.1	509.3 ± 8.1	22.1 ± 0.9 b	8.1 ± 1.0 a	30.2 ± 1.8 b
	Pungsannamul	212.7 ± 4.7	501.0 ± 16.4	37.1 ± 1.8 a	8.6 ± 3.2 a	45.7 ± 4.5 a
	Daewon	204.0 ± 4.4	393.3 ± 19.6	20.0 ± 3.8 b	5.4 ± 0.6 a	25.4 ± 3.4 b
	Seonpung	216.7 ± 14.7	460.0 ± 47.0	24.2 ± 4.6 b	4.5 ± 1.4 a	28.7 ± 5.1 b
	Seoritae	206.7 ± 5.7	414.0 ± 40.3	18.8 ± 5.0 b	6.8 ± 3.0 a	25.6 ± 7.4 b
	*p* value			0.0003	0.199	0.0009
	F value			11.93	1.75	9.01
	df			5	5	5

Values represent the mean ± SD. Means followed by the same letters in a column are not significantly different among soybean cultivars (ANOVA, Tukey’s HSD test, *p* < 0.05).

## Data Availability

The data supporting reported results can be found in the manuscript.

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
