# Peer review of "Feeding Behavior Comparison of Bean Bugs, Riptortus pedestris and Halyomorpha halys on Different Soybean Cultivars"

_insects, 2023, doi:10.3390/insects14040322_

Round 1

Reviewer 1 Report

Summary of work presented:

This study records EPG of two types of stink bug and the respective feeding habits of these stink bugs on six commercially relevant types of soybeans in Korea. The study examines the duration of piercing-feeding insects during different stages of feeding on soybeans for the two stink bugs (e.g. salivation and sap feeding, and xylem feeding). A survey of hemipterans collected in soybean fields for three months in 2021 was collected and reported. The two most prevalent species of hemipterans were R. pedestris and H. Halys—this is likely the reason the authors selected these two species of Hemiptera as the study species for the paper. Lastly, seed damage was assessed for the six different types of soybeans after the three-month period. The method of EPG is not trivial and recording the feeding behaviors under field conditions is an admirable effort. This manuscript will benefit from addressing the concerns and comments in the sections below.

Larger overarching notes:

-       The paper would benefit from elaborating on the significance of the different feeding patterns regarding the comparison of stink bugs and cotton aphids (as mentioned in Lines 160-161).

-       It is apparent to me that the time duration of feeding generally highest for the non-feeding stage, followed by feeding on the xylem, followed by feeding on the sap, then followed by salivating. Do the authors have a suspicion as to why sap feeding is relatively lower when compared to the xylem feeding?

-       Interestingly, the H. halys stink bug recorded no time feeding on sap or salivating on the Daepung-2ho and Seoritae soybeans. This is briefly mentioned in Line 193. I recommend that the authors elaborate on this observation.

-       The results/discussion should expand upon the significance of the findings, particularly regarding existing literature in the field and the more general IPM implications.

-       Lines 228-230- These sentences will require further elaboration.

Minor notes and corrections:

-       Line 98- I believe the authors meant a “Faraday” cage.

-       Figure 1 could benefit from increasing the image quality or resolution, increasing the font sizes, and labeling the axes which are missing labels.

-       Figure 2a- Typo on title. “Waveform” is missing an “m.”

-       Figure 2 caption- The caption can benefit from simply reiterating what the different waveforms correspond to (i.e. “Panel b is the E1 waveform (Salivation), Panel c is the E2 waveform (Sap feeding phase) and Panel d is the G waveform (Xylem feeding phase).”

-       Line 17- I believe the authors meant to refer to Table 2.

-       Line 114- When does the seed damage measurement and analysis take place? Is this after the grain filling period?

-       Lines 214-216 – The most and least damage rate (B+C) that the authors are referring to in this sentence appears to only include Site B. Is there a reason for only examining this site? Can the authors indicate that they are referring explicitly to Site B?

-       Table 3- In the titles of each column, the authors are using “phage” when I believe they mean to use “phase.”

Author Response

We appreciate for the consideration of three anonymous reviewer and final correction. All comments of reviewer on the manuscript were accepted and all comments are changed properly.

Thank you so much for your kind comments.

Ms. Ref. No. insects-2169882

Title: Feeding Behavior and Infestation of Bean Bugs, Riptortus pedestris and Halyomorpha halys, on Different Soybean Cultivars: An Electrical Penetration Graph Analysis

Reviewers' comments

Reviewer: 1

Comments to the Author

This study records EPG of two types of stink bug and the respective feeding habits of these stink bugs on six commercially relevant types of soybeans in Korea. The study examines the duration of piercing-feeding insects during different stages of feeding on soybeans for the two stink bugs (e.g. salivation and sap feeding, and xylem feeding). A survey of hemipterans collected in soybean fields for three months in 2021 was collected and reported. The two most prevalent species of hemipterans were R. pedestris and H. Halys—this is likely the reason the authors selected these two species of Hemiptera as the study species for the paper. Lastly, seed damage was assessed for the six different types of soybeans after the three-month period. The method of EPG is not trivial and recording the feeding behaviors under field conditions is an admirable effort. This manuscript will benefit from addressing the concerns and comments in the sections below.

Larger overarching notes:

- The paper would benefit from elaborating on the significance of the different feeding patterns regarding the comparison of stink bugs and cotton aphids (as mentioned in Lines 160-161).

Added.

- It is apparent to me that the time duration of feeding generally highest for the non-feeding stage, followed by feeding on the xylem, followed by feeding on the sap, then followed by salivating. Do the authors have a suspicion as to why sap feeding is relatively lower when compared to the xylem feeding?

 Analyzing the total duration of G, E1, and E2 waveform, except for Dp2 and Srt, they feed more xylem but feed similarly to sap. In the case of feeding only the xylem, it seems that the stink bug did not eat the phloem because it is not a preferred the soybean variety. It is possible that the longer the experiment time, the higher the sap feeding rate. However, since the EPG experiment was conducted with highly active insects, they may have been under a lot of stress. Interestingly, there was a slight difference between R. pedestris and H. halys.

- Interestingly, the H. halys stink bug recorded no time feeding on sap or salivating on the Daepung-2ho and Seoritae soybeans. This is briefly mentioned in Line 193. I recommend that the authors elaborate on this observation.

 Added.

- The results/discussion should expand upon the significance of the findings, particularly regarding existing literature in the field and the more general IPM implications.

Partially corrected.

- Lines 228-230- These sentences will require further elaboration.

 Corrected.

Minor notes and corrections:

- Line 98- I believe the authors meant a “Faraday” cage.

Corrected.

- Figure 1 could benefit from increasing the image quality or resolution, increasing the font sizes, and labeling the axes which are missing labels.

Corrected.

- Figure 2a- Typo on title. “Waveform” is missing an “m.”

Corrected.

- Figure 2 caption- The caption can benefit from simply reiterating what the different waveforms correspond to (i.e. “Panel b is the E1 waveform (Salivation), Panel c is the E2 waveform (Sap feeding phase) and Panel d is the G waveform (Xylem feeding phase).”

Corrected.

- Line 17- I believe the authors meant to refer to Table 2.

Yes. You're right.

- Line 114- When does the seed damage measurement and analysis take place? Is this after the grain filling period?

After the stink bug outbreak was investigated, all the soybean seeds were harvested and the damage rate was measured (= after the grain filling period).

- Lines 214-216 – The most and least damage rate (B+C) that the authors are referring to in this sentence appears to only include Site B. Is there a reason for only examining this site? Can the authors indicate that they are referring explicitly to Site B?

We compared and explained the damage rates of each soybean variety at sites A and B.

- Table 3- In the titles of each column, the authors are using “phage” when I believe they mean to use “phase.”

Corrected.

Thank you very much for your comments.

Reviewer 2 Report

Comments in the attached file.

Author Response

We appreciate for the consideration of three anonymous reviewer and final correction. All comments of reviewer on the manuscript were accepted and all comments are changed properly.

Thank you so much for your kind comments.

Reviewer 3 Report

Comments and Suggestions for Authors

The manuscript ‘Feeding Behavior and Infestation of Bean Bugs, Riptortus pedestris and Halyomorpha halys, on Different Soybean Cultivars: An Electrical Penetration Graph Analysis’ disclosing EPG analysis of two bean bugs on six soybean cultivars. The manuscript focuses on an interesting and hot topic in this field; however, it is still immature. More background of the two bean bugs need to be supplemented in the manuscript. The list of the corrected points described below.

LINE 9-22  The abstract of this article is a bit weak and the details should be provided.

Line 45  Riptortus pedestris is a major pest of leguminous plants in East Asia. The latest research progress cannot be ignored in the Introduction. It is better to quote relevant references to illustrate its importance.

Field cage assessment of feeding damage by Halyomorpha halys on kiwifruit orchards in China. Journal of Pest Science. DOI: 10.1007/s10340-020-01216-8

Laboratory evaluation of leguminous plants for the development and reproduction of the bean bug Riptortus pedestris (Hemiptera: Alydidae). Entomological Science.  https://doi.org/10.1111/ens.12525

Feeding of Riptortus pedestris on soybean plants, the primary cause of soybean staygreen syndrome in the Huang-Huai-Hai river basin. The Crop Journal.  https://doi.org/10.1016/j.cj.2018.07.008

Transgenerational changes in pod maturation phenology and seed traits of Glycine soja infested by the bean bug Riptortus pedestris. PLoS ONE.  https://doi.org/10.1371/journal.pone.0263904

Transcriptional profiling reveals a critical role for GmFT2a in soybean staygreen syndrome caused by the pest Riptortus pedestris. New Phytologist. DOI: 10.1111/NPH.18628

Line 46-48  ‘Large populations...in Korea and Japan.’ Please describe the location.

Line 64-65  EPG analysis of Riptortus pedestris and Halyomorpha halys has been reported. Please describe the novelty or innovation of this manuscript.

Characterization of electropenetrography waveforms for the invasive heteropteran pest, Halyomorpha halys, on Vicia faba leaves. Arthropod - Plant Interactions. DOI: 10.1007/s11829-019-09722-y

Feeding Behavior of Riptortus pedestris (Fabricius) on Soybean: Electrical Penetration Graph Analysis and Histological Investigations. Insects.  DOI: 10.3390/insects13060511

Line 88  The first-to-third-instar nymphs of R. pedestris are difficult to attach to the metal wire due to small size. How could you attach the second instar nymphs?

Line 89  Is there any obvious rhythm in the sucking or feeding behavior of the second instar nymphs in a day? Does this behavior affect EPG results? I think soybean secondary metabolites and pod structure are important factors affecting feeding, not population number in field.

Line 95  R. pedestris and H. halys prefer to feed on soybean pods, and which soybean growing stage you have used?

Line 103  Please revise the method to increase readability.

Line 108  One plant represent one copy ,right? Have you caged the plants? otherwise, it is hard to judge if it have been damaged by other stink bugs, and because you surveyed per week. your theme is to survey the damage of R. pedestris and H. halys. Also, you should have the control which promise no insects feeding.

Line 109  What method did you use to conduct your survey?

Line 118  the seeds was visually inspectedunder microscope?

Line 122-123  As far as I know, it is very likely that the soybean pods damaged by the bee border bugs will show this situation, but you have ruled out.

Line 142  The authors should provide sufficient evidence that differences in characteristics of EPG waveform among six soybean cultivars. Figure 1 should preferably be replaced with higher resolution figures. It is better to divide two species into two graphs, rather than one general graph.

Line 158  It's better to compare it with other bug (Pentatomidae, Alydidae or Coreidae).

Line 197.  Is there any correlation between the hemipterans collected in the field and the results of EPG analysis? The authors should provide sufficient evidence.

Line 225-227  In my opinion, the longer feeding time of stink bugs and the higher damage rate of soybeans did not necessarily indicate the preference of stink bugs. In the EPG experiment, it may be that the nutritional level of soybean could not be satisfied, so it needs to be fed for a long time. In the field experiment, it may be that soybeans are less resistant and more responsive to feeding, or because of long time feeding. Because your lacking the data that there were more stink bugs be found in pungsannamul (Bean cultivar) block. Maybe you are right, but i think you need more eviedence.

Line 230  I think that the discussion is too general and the understanding of the results obtained in this study is not deepened. For example, the authors did not discuss about soybean varieties, these statements should be kept to a maximum as their direct impact on the results is unclear. Please revise the discussion based on the EPG analysis of known Hemiptera insects to be deep as a whole.

Author Response

We appreciate for the consideration of three anonymous reviewer and final correction. All comments of reviewer on the manuscript were accepted and all comments are changed properly.

Thank you so much for your kind comments.

Ms. Ref. No. insects-2169882

Title: Feeding Behavior and Infestation of Bean Bugs, Riptortus pedestris and Halyomorpha halys, on Different Soybean Cultivars: An Electrical Penetration Graph Analysis

Reviewers' comments

Reviewer: 3

The manuscript ‘Feeding Behavior and Infestation of Bean Bugs, Riptortus pedestris and Halyomorpha halys, on Different Soybean Cultivars: An Electrical Penetration Graph Analysis’ disclosing EPG analysis of two bean bugs on six soybean cultivars. The manuscript focuses on an interesting and hot topic in this field; however, it is still immature. More background of the two bean bugs need to be supplemented in the manuscript. The list of the corrected points described below.

LINE 9-22 The abstract of this article is a bit weak and the details should be provided.

 Corrected.

Line 45 Riptortus pedestris is a major pest of leguminous plants in East Asia. The latest research progress cannot be ignored in the Introduction. It is better to quote relevant references to illustrate its importance.

Corrected.

Field cage assessment of feeding damage by Halyomorpha halys on kiwifruit orchards in China. Journal of Pest Science. DOI: 10.1007/s10340-020-01216-8

Laboratory evaluation of leguminous plants for the development and reproduction of the bean bug Riptortus pedestris (Hemiptera: Alydidae). Entomological Science.  https://doi.org/10.1111/ens.12525

Feeding of Riptortus pedestris on soybean plants, the primary cause of soybean staygreen syndrome in the Huang-Huai-Hai river basin. The Crop Journal.  https://doi.org/10.1016/j.cj.2018.07.008

Transgenerational changes in pod maturation phenology and seed traits of Glycine soja infested by the bean bug Riptortus pedestris. PLoS ONE.  https://doi.org/10.1371/journal.pone.0263904

Transcriptional profiling reveals a critical role for GmFT2a in soybean staygreen syndrome caused by the pest Riptortus pedestris. New Phytologist. DOI: 10.1111/NPH.18628

Line 46-48 ‘Large populations...in Korea and Japan.’ Please describe the location.

It is being observed almost nationwide.

Line 64-65 EPG analysis of Riptortus pedestris and Halyomorpha halys has been reported. Please describe the novelty or innovation of this manuscript.

 This experiment compared the feeding behavior of Riptortus pedestris and Halyomorpha halys in 6 types of soybeans widely cultivated in Korea using the EPG technique. A similar explanation is given in the last paragraph of the introduction.

Characterization of electropenetrography waveforms for the invasive heteropteran pest, Halyomorpha halys, on Vicia faba leaves. Arthropod - Plant Interactions. DOI: 10.1007/s11829-019-09722-y

Feeding Behavior of Riptortus pedestris (Fabricius) on Soybean: Electrical Penetration Graph Analysis and Histological Investigations. Insects.  DOI: 10.3390/insects13060511

Line 88 The first-to-third-instar nymphs of R. pedestris are difficult to attach to the metal wire due to small size. How could you attach the second instar nymphs?

In our laboratory, we have been experimenting with EPG analysis since 2006, and we have dealt with many insects such as Myzus persicae, Aphis gossypii, Sogatella furcifera, Nilaparvata lugens, and Bemisia tabaci, etc., so we can handle them easily.

Line 89 Is there any obvious rhythm in the sucking or feeding behavior of the second instar nymphs in a day? Does this behavior affect EPG results? I think soybean secondary metabolites and pod structure are important factors affecting feeding, not population number in field.

I think this is a reliable result because 49 - 71 repetitions were done for each variety for 6 hours.

Line 95 R. pedestris and H. halys prefer to feed on soybean pods, and which soybean growing stage you have used?

We used soybean seedlings of six cultivars (Daepung-2ho, Daechan, Pungsannamul, Daewon, Seonpung, and Seoritae) approximately 10-14 days after planting.

Line 103 Please revise the method to increase readability.

Corrected.

Line 108 One plant represent one copy, right? Have you caged the plants? otherwise, it is hard to judge if it have been damaged by other stink bugs, and because you surveyed per week. your theme is to survey the damage of R. pedestris and H. halys. Also, you should have the control which promise no insects feeding.

Each experimental group was covered with a net to prevent stink bugs from escaping.

Line 109 What method did you use to conduct your survey?

 We walked slowly through the soybean fields, “searching and netting” the insect pests (added in line 109).

Line 118 the seeds was visually inspected under microscope?

 First, the damage rate of beans was observed with the naked eye, and the invisible ones were observed under a microscope.

Line 122-123 As far as I know, it is very likely that the soybean pods damaged by the bee border bugs will show this situation, but you have ruled out.

Yes. We excluded the areas we tested because the bees suffered little damage.

Line 142 The authors should provide sufficient evidence that differences in characteristics of EPG waveform among six soybean cultivars. Figure 1 should preferably be replaced with higher resolution figures. It is better to divide two species into two graphs, rather than one general graph.

Corrected.

Line 158 It's better to compare it with other bug (Pentatomidae, Alydidae or Coreidae).

Corrected.

Line 197. Is there any correlation between the hemipterans collected in the field and the results of EPG analysis? The authors should provide sufficient evidence.

First, in the EPG results of Table 2, the feeding time was short in Daepung-2ho and long in Pungsannamul. In the damage rate in the soybean field in Table 4, similar to the results in Table 2, the damage rate was the lowest in Daepung-2ho and the highest in Pungsannamul. However, as your opinion, the longer feeding time of stink bugs and the higher damage rate of soybeans did not necessarily indicate the preference of stink bugs.

Line 225-227 In my opinion, the longer feeding time of stink bugs and the higher damage rate of soybeans did not necessarily indicate the preference of stink bugs. In the EPG experiment, it may be that the nutritional level of soybean could not be satisfied, so it needs to be fed for a long time. In the field experiment, it may be that soybeans are less resistant and more responsive to feeding, or because of long time feeding. Because your lacking the data that there were more stink bugs be found in pungsannamul (Bean cultivar) block. Maybe you are right, but i think you need more eviedence.

You are right. So, I quoted this in the discussion section. The quality of discussion seems to be much better. If you object, I will delete it.

Line 230 I think that the discussion is too general and the understanding of the results obtained in this study is not deepened. For example, the authors did not discuss about soybean varieties, these statements should be kept to a maximum as their direct impact on the results is unclear. Please revise the discussion based on the EPG analysis of known Hemiptera insects to be deep as a whole.

Partially corrected.

Thank you very much for your comments.

Round 2

Reviewer 2 Report

Dear authors, I have just seen the current file and I saw that it has improved and, in my opinion, is suitable to be recommended for publication.
Best

Author Response

Thank you very much.

Reviewer 3 Report

The authors responded to the comments and made changes, but there is another inaccuracy in Line 135-137 (and D-type: seeds that are immature and have little shape. Because the cause of seed D-type could be stink bug damage or physiological disturbance, it was excluded from the damage analysis.) This sentence should be deleted.

Author Response

Thank you for your comment. We deleted it.
